# Unimodal-driven Distillation in Multimodal Emotion Recognition with Dynamic Fusion

## Abstract

Multimodal Emotion Recognition in Conversations (MERC) seeks to identify emotional states across multiple modalities, including text, audio, and video. This field of study is pivotal for advancing machine intelligence, with significant implications for applications such as intelligent dialogue systems and public opinion analysis. Most existing approaches primarily employ full-sequence interaction and distillation techniques, aiming to construct a comprehensive global contextual understanding while simultaneously enhancing the interaction among heterogeneous modalities. However, the presence of repetitive and redundant information, coupled with gradient conflicts arising from modal heterogeneity, can significantly impede the effectiveness of multimodal learning and long-range relationship modeling. In this work, we propose an innovative heterogeneous multimodal integration method called SUMMER, grounded in attention mechanism and knowledge distillation techniques, which facilitates dynamic interactive fusion of multimodal representations. Specifically, the Sparse Dynamic Mixture of Experts strategy is proposed to dynamically adjust the relevance of the temporal information to construct local to global token-wise interactions. Then a Global Mixture of Experts is employed to enhance the model's overall contextual understanding across modalities. Notably, we introduce retrograde distillation that utilizes a pre-trained unimodal teacher model to guide the learning of multimodal student model, intervening and supervising multimodal fusion within both the latent and logit spaces. Experiments on the IEMOCAP and MELD datasets demonstrate that our SUMMER framework consistently outperforms existing state-of-the-art methods, with particularly significant improvements in recognizing minority and semantically similar emotions in MERC tasks.

## 1 Introduction

Multimodal Emotion Recognition in Conversations (MERC) Poria et al. (2019) seeks to elucidate the emotional dynamics inherent in interactions, thereby enhancing human-computer interaction Cowie et al. (2001) and fostering empathy across diverse domains such as digital humans, healthcare Pujol et al. (2019), and social media analytics Andalibi & Buss (2020). Unlike traditional methods, multimodal emotion analysis integrates cues from text, audio, and visual modalities Zhang et al. (2024) which can capture nuanced emotional cues and facilitate corrective feedback mechanisms from varied contexts.

In MERC tasks, most existing studies focus on constructing global context understanding and improving cross-modal fusion. The RNN-based model DialogueRNN Majumder et al. (2019) leverages Recurrent Neural Networks to capture global temporal dependencies, while the GNN-based model CORECT uses Relational Temporal Graph Neural Networks to represent multimodal relationships. In contrast, the Transformer-based model MultiEMO and SDT Shi & Huang (2023); Ma et al. (2023) employs attention mechanisms to prioritize long-range dependencies, integrating contextual information across multiple modalities.

Despite advances in MERC, challenges such as inefficient modal association persist. As illustrated in Figure 1 (a): (1) In the 6th utterance, phrases like "but no" and "she's not my girlfriend" clearly indicate sadness. However, if the model overemphasizes earlier positive expressions like "we communicate on a daily," it may incorrectly classify the emotion as happiness. This underscores the

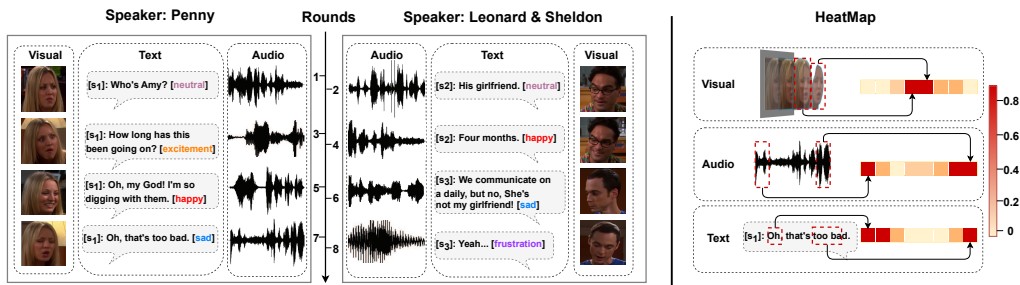

Figure 1: (a) A representative example of multimodal emotion recognition in conversations. For each given sentence, it contains three modal information about the speaker, text, video, and audio. The task of MERC is to identify the emotional labels contained in the three modal information. (b) Examples of the limitations of the traditional MoE model for MERC tasks.

risk of focusing on local context while neglecting key emotional cues. (2) In the 3th utterance, the correct label is "excited," but dynamic changes in facial expressions and vocal tone might mislead the model to classify it as anger. Such intense emotional variations can be misinterpreted as negative emotions, highlighting the complexity of multimodal data in emotion recognition tasks.

Although the Mixture of Experts (MoE) model dynamically selects the most suitable token experts via a gating mechanism, improving multimodal fusion and association efficiency, it also has limitations. As shown in Figure 1 (b), MoE only selects a fixed Top-K subset or weights all experts for reasoning, limiting its adaptability in complex MERC environments. Therefore, dynamically selecting token-level information and incorporating a global Mixture of Experts adapter is essential for optimizing contextual understanding and filtering redundant information.

To enhance the cross-modal fusion of heterogeneous modalities refined by MoE, the Transformer-based SDT model Ma et al. (2023) employs a self-distillation approach to guide multimodal fusion learning. Additionally, cross-modality distillation enhances fusion by enabling knowledge transfer between heterogeneous features. However, self-distillation methods often face gradient conflict issues, where gradients from the teacher and student model interfere during training. Resolving fusion disorientation is critical to improving the effectiveness of multimodal fusion.

In this work, we propose a **S**parse **U**nimodal-driven distillation for **M**ulti-**M**odal **E**motion **R**ecognition named *SUMMER* to enhance modal association learning and fusion disorientation. First, we employ Sparse Dynamic MoE (SDMoE) to enhance token-wise interaction for high-quality localized information and mitigate the impact of redundant data on cross-modal fusion. Then we introduce Hierarchical Cross-Modal Fusion (HCMF) with Global MoE (GMoE) to adaptively capture and unify intrinsic links between modalities to improve global context understanding. Additionally, we propose a novel Interactive Knowledge Distillation (IKD) where a high-performing unimodal teacher model guides the learning of a multimodal student model, facilitating directed learning and reducing gradient conflicts caused by modal discrepancies.

The main contributions of this work are summarized as follows:

- We propose a Sparse Dynamic Mixture of Experts and Hierarchical Cross-Modal Fusion method to enhance local key token selection and improve global context understanding, thereby refining heterogeneous modal information for more effective multimodal fusion.

- We introduce a retrograde distillation strategy where a unimodal-driven teacher model guides the multimodal student model, standardizing and addressing fusion disorientation in multimodal learning.

- Our model significantly outperforms state-of-the-art benchmarks on the IEMOCAP and MELD datasets, demonstrating superior performance in capturing subtle emotional nuances, and excelling in semantically similar and underrepresented emotion categories.

## 2 RELATED WORK

**Multimodal Emotion Recognition in Conversations.** The core objective of MERC Dashtipour et al. (2016) is to analyze speakers' emotional states by leveraging multimodal data over time. While early approaches relied heavily on GNN-based Ghosal et al. (2019); Song et al. (2023); Hu et al. (2021) and RNN-based architectures Poria et al. (2017); Majumder et al. (2019); Jiao et al. (2019); Li et al. (2022), which were standard in natural language processing, these recurrent models faced limitations in handling long sequences and lacked scalability. In contrast to these models, contemporary approaches aim to capture both intra- and inter-modal interactions, leading to more nuanced emotional analysis by unifying information from text, audio, and visual modalities. Techniques such as tensor fusion, as employed by LMF Liu et al. (2018), manage complementary information while reducing redundancy across modalities, further enhancing multimodal fusion. Additionally, MM-DFN Hu et al. (2022) dynamically captures contextual and multimodal features while minimizing irrelevant information across modalities.

**Transformer-based Models.** The introduction of Transformer models Vaswani (2017) revolutionized MERC by enabling efficient parallel computing and long-sequence modeling through self-attention mechanisms, leading to significant advancements in intra- and inter-modal fusion. Models like CTNet Lian et al. (2021) employ single and cross-modality Transformers, while CKETF Ghosh et al. (2021) enhances context and knowledge representation within a Transformer framework. TL-ERC leverages transfer learning to improve performance across tasks.

To improve multimodal understanding, dynamic attention mechanisms are employed to adjust attention weights, enabling more effective cross-modal encoding. TFR-Net Yuan et al. (2021) and Emocaps Li et al. (2022) leverage intra- and inter-modal attention to capture sentiment trends. Tailor Zhang et al. (2022) uses a Transformer-based unimodal extractor and a multi-label bootstrap decoder to model dependencies between labels and modalities. SDT Ma et al. (2023) introduces an Intra- and Inter-modal Transformer for emotional interactions across modalities and sessions, while TACFN Liu et al. (2023) proposes an Adaptive Inter-modal Fusion Network to reduce redundancy and improve feature integration.

**Knowledge Distillation.** Knowledge Distillation (KD) Gou et al. (2021) has become a powerful method for compressing models and improving efficiency by transferring knowledge from a larger teacher model to a smaller student model. In multimodal emotion recognition, KD enables the integration of complementary information across modalities, helping the student model capture richer emotional representations. SENet Albanie et al. (2018) transfers visual knowledge into speech emotion recognition models using unlabeled video data. Schoneveld Schoneveld et al. (2021) utilizes the KD method to improve the performance of models in facial expression recognition. Similarly KIAN Wang et al. (2020) proposes K-injection subnetworks to distill linguistic and acoustic information, allowing implicit knowledge transfer in audiovisual models for group emotion recognition.

The majority of these approaches rely on offline distillation, which necessitates the pre-training of a large teacher model to guide the learning of smaller student model. However, little attention has been given to using a smaller unimodal-driven teacher model to instruct more complex multimodal students, which has the potential for effective cross-modal learning. This gap serves as the primary motivation for our work.

## 3 METHODOLOGY

### 3.1 TASK DEFINITION

In MERC tasks, each conversation consists of $n$ utterances $\{u_1, u_2, ..., u_n\}$ and $m$ speakers $\{s_1, s_2, ..., s_m\}$. Each utterance $u_i$ comprises three modalities, represented as $u_i = \{u_i^t, u_i^a, u_i^v\}$, where $t$, $a$, and $v$ denote text, audio, and visual modalities, respectively. The objective is to predict the sentiment classification label $y_i$ corresponding to each $u_i$ within the conversation.

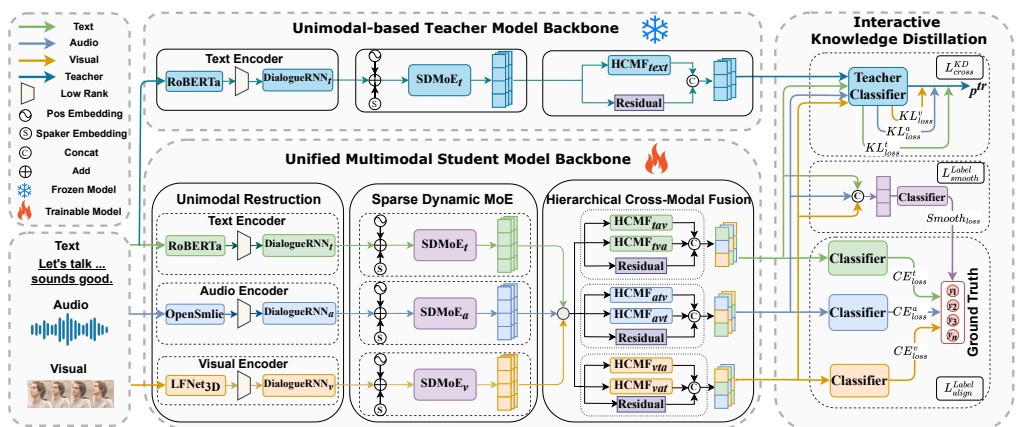

Figure 2: Illustration of the SUMMER framework, which comprises the Unimodal Teacher Model, Unified Multimodal Student Model, and Interactive Knowledge Distillation. The frozen teacher model is dedicated to mentoring the student model by providing a comprehensive guide for learning.

## 3.2 MODEL OVERVIEW

As shown in Figure 2, SUMMER consists of four core modules: Unimodal Reconstruction, Sparse Dynamic Mixture of Experts (SDMoE), Hierarchical Cross-Modal Fusion (HCMF), and Interactive Knowledge Distillation (IKD). The unimodal encoder extracts features from text, audio, and visual inputs, while SDMoE focuses on token-wise interaction, dynamically adjusting global context associations and filtering redundant information. HCMF enriches semantics by aligning multimodal weights, and IKD improves cross-modal feature fusion through efficient knowledge transfer, leveraging lightweight pre-trained teacher models via latent and logit spaces.

## 3.3 UNIMODAL RESTRUCTION

**Unimodal Encoder.** For the Text Encoder, we use the pre-trained roBERTa model to extract text features $h_i^t \in \mathbb{R}^{l_s \times d_t}$, incorporating speaker identity and dialogue separators to capture both intra- and inter-speaker context. The Audio Encoder leverages OpenSMILE to extract 6,373-dimensional acoustic features $h_i^a \in \mathbb{R}^{l_s \times d_a}$, which are reduced to 512 dimensions for efficiency. To address challenges with direct $\text{CNN}_{(3D)}$ video processing, we propose $\text{LFNet}_{3D}$ (see details in A.1) to produce 256-dimensional spatio-temporal features $h_i^v \in \mathbb{R}^{l_s \times d_v}$. Finally, DialogueRNN is employed to capture global emotional trends and speaker-emotion dynamics in conversations.

**Utterance-Speaker Embeddings.** As illustrated in the 4-7th utterances in Figure 1 (a), emotion of the current speaker directly influences the next speaker. To effectively model the relationships between speaker identity $S_j$ and utterance in affective states, it is crucial to incorporate a latent speaker representation into the positional embeddings. This is achieved using an input feature set $H_i^m = \{h_i^t, h_i^a, h_i^v\}$, which includes text, audio, and visual features extracted by a unimodal encoder.

$$S_j = V_{s_j} o_{s_j} \in \mathbb{R}^{ls \times d_s}, \tag{1}$$

$$U_e = H_i^m + S_i + P_i, \tag{2}$$

where $j$ represents the identity of different speakers, $V_{s_j}$ is a learnable speaker identity embedding, $o_{s_j}$ is the one-hot encoding of each speaker, and $P_i$ represents the absolute position embeddings of the utterance.

## 3.4 SPARSE DYNAMIC MIXTURE OF EXPERTS

In certain cases, the complexity of the dialogue environment impacts model accuracy. To mitigate this, we propose a SDMoE module, as shown in Figure 3 (a), comprising three key components: an Auxiliary Expert Network, a Dynamic Routing Mechanism, and a Global MoE.

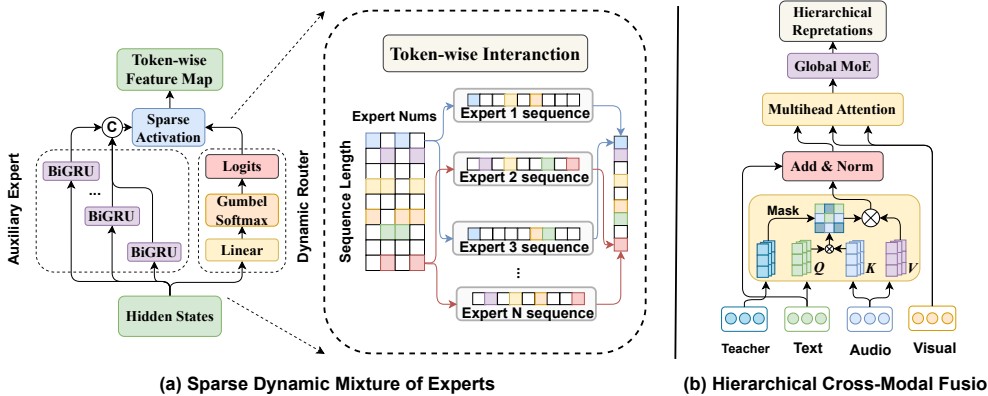

**(a) Sparse Dynamic Mixture of Experts**  **(b) Hierarchical Cross-Modal Fusion**

Figure 3: (a) SDMoE comprises two main components: the Auxiliary Expert Network and the Dynamic Routing Mechanism. Specifically, the dynamic router adjusts the relevance of the attention map to facilitate local token-wise interactions. (b) HCMF integrates a Teacher-guided Cross-Modal Fusion with a GMoE module to enhance overall contextual understanding across modalities.

**Auxiliary Expert Network.** We capture modality-specific emotional semantics at multiple levels using a set of BiGRU experts. Each expert model processes the encoded features $E_i = \text{BiGRU}(U_e)$, enhancing the model's ability to adapt to temporal dependencies while mitigating noise and redundancy. Parameters are shared within intra-modal components but remain independent across inter-modal components. The expert network outputs are aggregated as $E_o = \{E_{o1}, E_{o2}, \ldots, E_{on}\}$, where $n$ denotes the number of experts.

**Dynamic Routing Mechanism.** Instead of summing the weights of all or Top-K expert models as in traditional MoE, we propose a dynamic routing mechanism $G_{dyn}$, which dynamically adjusts the number of experts according to the simplicity of the scenario. The gating network generates a global representation of the multimodal context and produces a sparse key representation $M_{sparse}$.

$$G_{dyn} = \begin{cases} \frac{Softmax(W_g)}{T}, & if \ \ W_g \in (\mu - 2\sigma, \mu + 2\sigma) \\ 0, & otherwise \end{cases} \tag{3}$$

where $T$ represents a temperature-adjusted parameter to control weight distribution, while $\mu$ and $\sigma$ denote the mean and standard deviation of the weights, respectively. Weights in $W_g$ are selectively deactivated if they fall outside the range $(\mu - 2\sigma, \mu + 2\sigma)$, with non-critical features set to zero.

However, our selection mechanism involves a discrete sampling process, resulting in a non-differentiable model during gradient propagation. To address this, we introduce Gumbel noise to ensure differentiability during backpropagation, $g_{noise} = -log(-log(R_i))$, where $R_i$ is a random variable sampled from a uniform distribution $(0, 1)$. The improved $\hat{G_{dyn}}$ can be expressed as:

$$\hat{G_{dyn}} = \frac{\exp(\frac{W_g + g_{noise}}{\tau})}{\sum_1^n \exp(\frac{W_g + g_{noise}}{\tau})}, \tag{4}$$

$$M_{sparse} = \sum_i^n (\hat{G_{dyn}} \times E_o), \tag{5}$$

where $\tau$ is a learnable parameter that controls the smoothness of the distribution.

**Global MoE.** To mitigate the potential loss of global contributions from various modules caused by directly using decision variables for inference, we introduced a GMoE that dynamically selects representations $H_m$ from the HCMF 3.5 modules, managed by a global router. Assuming the expert outputs are $F_o = \{f_t, f_a, f_v\}$, the global router is defined as $G_{global} = V_g \times F_o$, where $V_g$ is a learnable global dynamic adapter. The multimodal decision vector $H_{fuse}$ can then be computed as:

$$H_{fuse} = \sum_i^n (V_g \times F_o \times H_m). \tag{6}$$

Notably, leveraging the sparse dynamic routing mechanism extends the capacity of the global router without significantly increasing training or inference time. And our proposed GMoE can be applied to any layer for intermediate output processing.

### 3.5 HIERARCHICAL CROSS-MODAL FUSION

**Unimodal-driven Multimodal Learning.**   Modal imbalance often occurs in multimodal learning when the model fails to effectively leverage all modalities, resulting in unstable performance. Therefore, we designed and pre-trained a unimodal teacher model using the SDMoE module. Experimental results (discussed in Section 4.5) demonstrate that the text-based teacher model achieves the best performance in a unimodal setup. This finding motivates the use of a single modality as a prior to bootstrap cross-modal feature distillation and enables transfer learning.

**Teacher-Guided Cross-Modal Fusion.**   Our proposed teacher-guided HCMF framework (Figure 3 (b)), consists of three sub-branches: $HCMF_t$, $HCMF_a$, and $HCMF_v$, each employing a BERT-like bidirectional encoder to process text, audio, and visual modalities, respectively. Taking the $HCMF_t$ branch as an example, we define the inputs to the student model as $Qst^t$, $K_{st}^t$, $V_{st}^t \in \mathbb{R}^{l_m \times d_m}$, while the intermediate outputs of the teacher model are denoted as $Q_{tr}$, $K_{tr}$, $V_{tr}$. To transfer the intermediate features from the teacher model to the student model via masking, a mask $M_{ij}$ is applied to ensure that the student model's intermediate representations align with the teacher model's guidance. The teacher-guided attention $DynAttn$ can be described as follows:

$$M_{ij} = \begin{cases} 1, & if \quad \sqrt{(s_{tr} - s_{st})^2} > 0.5, \\ 0, & otherwise, \end{cases} \tag{7}$$

$$DynAttn = \sum_{i=1}^{n}((1 - \phi M_{ij}) \cdot Softmax(\frac{Q_{st}^t K_{st}^{t^T}}{\sqrt{d}})V_{st}^t), \tag{8}$$

where $s_{tr}$ and $s_{st}$ represent the dot products of the $Q$ (query) and $K$ (key) matrices from the teacher and student model, respectively. $\phi$ is a dynamic adjustment factor that moderates the masking field. Based on $DynAttn$, we can fuse multiple modals dynamically, where $H_{ta}$ (text-audio) and $H_{tav}$ (text-audio-visual) are the hierarchical cross-modal fusion outputs, formally defined as follows:

$$H_{ta} = DynAttn_{ta}(Q_{st}^t, K_{st}^a, V_{st}^a), \tag{9}$$

$$H_{tav} = DynAttn_{tav}(H_{ta}, K_{st}^v, V_{st}^v). \tag{10}$$

For encoder at the same level, intra-modal interaction occurs via multi-head attention which enhances high-level semantic fusion. To ensure smoother cross-modal fusion, residual blocks are introduced to retain more original modal information.

### 3.6 INTERACTIVE KNOWLEDGE DISTILLATION

Verified by previous work on the distillation method Wang et al. (2024), relying solely on the teacher's final representations can lead to gradient conflicts due to the use of hard labels. Our proposed IKD approach (Figure 2) updates the student model's intermediate parameters by transferring knowledge in the space of homogeneous probability distributions for heterogeneous modal features, effectively mitigating prediction bias through the use of soft labeling.

**Interactive KD.**   To transfer knowledge without making the student model overly reliant on the teacher, we freeze the teacher's parameters and apply its classifiers to the student's intermediate features. This approach ensures that the heterogeneous modal features are mapped into a uniform distribution space. Simultaneously, we constrain the labels of the student model and supervise the feature fusion by leveraging the gaps in the logit space. The interaction loss $L_{cls}^{KD}$ is computed using KL divergence and Cross Entropy loss, which is defined as:

$$L_{cross}^{KD} = \sum_{i=1}^{N} \hat{p_{m_i}} \log \frac{\hat{p_{m_i}}}{\bar{p_{m_i}}}, \tag{11}$$

$$L_{align}^{Label} = -\sum_{i=1}^{N} \sum_{j=1}^{C} gt_i \, log(\hat{p_{m_i}}), \tag{12}$$

where $\hat{p_{m_i}}$ and $p_{m_i}^-$ represent the predicted distributions of the student and teacher intermediate features, both processed through the teacher model's classifier. $gt_i$ is the ground truth label.

**Inner KD.** Intermediate feature knowledge is transferred from the teacher to the student model, allowing the student $f_{m_i}^{st}$ to replicate the teacher's feature distribution $f_{m_i}^{tr}$. The discrepancy between their feature distributions is measured using MSE loss. Inner loss $L_{inner}^{KD}$ can be represent as follow:

$$L_{inner}^{KD} = \sum_{i=1}^{N} \sum_{j=1}^{C} ||f_{m_i}^{tr} - f_{m_i}^{st}||_2, \tag{13}$$

**Label Smooth Loss.** To reduce sensitivity to noise and prevent overconfidence in single categories, we employ soft labels instead of hard labels. This adjustment mitigates the risk of excessive reliance on incorrect teacher predictions. The corresponding smooth loss function is defined as:

$$L_{smooth}^{Label} = -\sum_{j=1}^{C} \left( \frac{\exp(p_{m_i}^{st})}{\sum_i^N \exp(p_{m_i}^{st})} \cdot \delta(gt_i) \right), \tag{14}$$

where $C$ represents the number of categories, $p_{m_i}$ is the prediction of multimodal fusion vector pass through the student model's classifier, $gt$ denotes the target label. For the correct category $gt_i = \epsilon$, while for the other categories $gt_j = (1-\epsilon)/(C-1)$ where $\epsilon \in (0,1)$.

**Training Objectives.** Our overall training objective of Interactive KD can be represented below, where $\kappa_1, \kappa_2, \kappa_3$ are compromise parameters between different objectives. In particular, $\kappa_4$ is set with a higher weight to minimize the impact of teacher model bias on the student model.

$$L_{KD} = \kappa_1 L_{cross}^{KD} + \kappa_2 L_{align}^{Label} + \kappa_3 L_{inner}^{KD} + \kappa_4 L_{smooth}^{Label}. \tag{15}$$

## 4 EXPERIMENTAL SETTINGS

### 4.1 DATASETS AND EVALUATION METRICS

To verify the validity of our proposed SUMMER model, we perform experiments on two widely-used MERC datasets, IEMOCAP Busso et al. (2008) and MELD Poria et al. (2018), which consist of multimodal data (text, audio, and video). IEMOCAP comprises 12 hours of conversations annotated with six emotion labels, while MELD contains dialogue clips from the TV show Friends with seven distinct emotion labels. In our experiments, we report accuracy (Acc) and F1-score for each emotion category, along with the overall weighted average accuracy (w-Acc) and weighted average F1 (w-F1) to compare the performance of the proposed method against baseline approaches.

### 4.2 BASELINES

We compare our model against several strong baselines: DialogueRNN Majumder et al. (2019) uses GRUs to model speaker states, context, and emotions, while DialogueGCN Ghosal et al. (2019) applies GCNs to represent conversations as graphs. MMGCN Hu et al. (2021) and CORECT Nguyen et al. (2023) use GCNs with dynamic fusion for multimodal context modeling, and MultiEMO Shi & Huang (2023) employs correlation-aware attention for multimodal fusion. SDT Ma et al. (2023) leverages self-distillation to capture intra- and inter-modal interactions, and CHFusion Majumder et al. (2018) introduces a hierarchical fusion strategy for restructuring contextual information.

### 4.3 IMPLEMENTATION DETAILS

We implemented the model in PyTorch, using the Adam optimizer with learning rates of 1e-4 for IEMOCAP and 5e-5 for MELD, with batch sizes of 32 and 100, respectively. Input dimensions are 100 for text and audio, and 256 for visual features in IEMOCAP, while MELD uses 768 for text, 512 for audio, and 1000 for visual inputs. The HCMF architecture includes a hidden size of 1024, 4 attention heads, and 6 cross-modal fusion layers, with L2 weight decay set to 1e-5.

Table 1: Quantative comparisons on IEMOCAP(6-ways) multimodal (A+V+T) setting.

| Models | happy | | sad | | neutral | | anger | | excitement | | frustration | | w-ACC | w-F1 |
|---|---|---|---|---|---|---|---|---|---|---|---|---|---|---|
| | ACC | F1 | ACC | F1 | ACC | F1 | ACC | F1 | ACC | F1 | ACC | F1 | | |
| DialogueRNN | 44.05 | 32.46 | 86.61 | 82.73 | 54.08 | 54.64 | 67.72 | 65.24 | 63.71 | 70.64 | 56.23 | 57.11 | 61.81 | 61.55 |
| DialogueGCN | 61.11 | 51.87 | 84.90 | 76.76 | 69.27 | 56.76 | 76.47 | 62.26 | 76.25 | 72.71 | 50.39 | 58.04 | 69.73 | 63.07 |
| MMGCN | 48.94 | 38.66 | 80.54 | 76.39 | 59.56 | 61.73 | 74.68 | 68.18 | 71.91 | 74.80 | 60.53 | 62.97 | 65.87 | 65.67 |
| CORECT | 59.15 | 58.74 | 86.18 | 80.95 | 71.43 | 69.52 | 63.74 | 65.91 | 80.60 | 76.19 | 62.89 | 68.11 | 71.44 | 70.81 |
| MultiEMO | 53.80 | 56.29 | 83.95 | 80.18 | 75.84 | 69.76 | 67.86 | 67.46 | 79.78 | 76.01 | 64.40 | 69.42 | 72.31 | 71.64 |
| SDT | 61.96 | 65.80 | 85.46 | 82.20 | 76.16 | 72.70 | 63.27 | 67.76 | 78.12 | 82.94 | 64.51 | 67.90 | 74.44 | 74.13 |
| CHFusion | - | - | - | - | - | - | - | - | - | - | - | - | 76.50 | 76.80 |
| Teacher Model | 70.83 | 73.12 | 82.79 | 83.61 | 84.86 | 74.23 | 65.22 | 71.95 | 82.94 | 81.30 | 68.63 | 70.10 | 75.21 | 74.22 |
| Student Model | 71.72 | 74.29 | 82.52 | 85.47 | 78.45 | 80.46 | 75.97 | 72.67 | 88.76 | 84.34 | 73.94 | 73.42 | 79.11 | 78.95 |

Table 2: Quantative comparisons on MELD(7-ways) multimodal (A+V+T) setting.

| Models | neutral | | surprise | | fear | | sadness | | joy | | disgust | | anger | | w-ACC↑ | w-F1↑ |
|---|---|---|---|---|---|---|---|---|---|---|---|---|---|---|---|---|
| | ACC | F1 | ACC | F1 | ACC | F1 | ACC | F1 | ACC | F1 | ACC | F1 | ACC | F1 | | |
| MMGCN | 68.87 | 77.51 | 48.12 | 46.80 | 0 | 0 | 50.00 | 13.33 | 55.46 | 51.47 | 0 | 0 | 45.40 | 45.60 | 56.85 | 57.35 |
| DialogueRNN | 71.62 | 75.66 | 52.17 | 46.97 | 0 | 0 | 32.46 | 22.98 | 48.00 | 52.00 | 0 | 0 | 43.60 | 45.88 | 55.83 | 57.37 |
| DialogueGCN | 79.06 | 75.80 | 53.02 | 50.42 | 0 | 0 | 17.79 | 23.72 | 59.20 | 55.48 | 0 | 0 | 50.43 | 48.27 | 60.96 | 58.72 |
| CORECT | 80.00 | 81.60 | 58.49 | 49.60 | 37.90 | 26.47 | 52.53 | 43.78 | 67.79 | 63.32 | 44.83 | 31.58 | 52.72 | 51.64 | 66.01 | 65.92 |
| SDT | 76.96 | 79.85 | 56.75 | 57.54 | 25.00 | 17.95 | 58.20 | 43.03 | 65.72 | 64.56 | 39.47 | 28.30 | 50.64 | 53.80 | 66.10 | 66.19 |
| MultiEMO | 78.55 | 79.94 | 54.49 | 58.28 | 36.00 | 24.00 | 56.15 | 43.20 | 61.06 | 64.64 | 43.75 | 28.00 | 53.31 | 53.47 | 66.43 | 66.40 |
| Teacher Model | 82.78 | 76.92 | 62.70 | 65.35 | 52.80 | 55.74 | 49.37 | 45.66 | 65.13 | 69.03 | 45.37 | 45.04 | 52.44 | 56.59 | 66.92 | 67.59 |
| Student Model | 86.29 | 83.44 | 62.66 | 68.95 | 53.42 | 56.39 | 49.38 | 43.04 | 66.86 | 70.96 | 45.28 | 47.52 | 55.13 | 57.33 | 68.78 | 69.81 |

## 4.4 Results and Analysis

Tables 1 and 2 represent a comparative analysis of performance metrics for the baseline models on the IEMOCAP and MELD datasets.

On the IEMOCAP dataset, the proposed SUMMER framework achieves a 2.61% improvement in w-ACC and 2.15% in w-F1, surpassing baselines like CHFusion, particularly in minority classes such as "excitement." The teacher model also outperforms prior approaches, with notable gains of 9.76% in w-ACC and 8.49% in w-F1 for the "happy" category. Improvements in "sadness" (1.86%) and "frustration" (3.32%) further demonstrate the effectiveness of token-wise interaction and soft-labeling in differentiating similar emotions.

On the MELD dataset, the teacher model surpasses all existing models in overall w-ACC and w-F1. The student model demonstrates strong performance in recognizing underrepresented emotions, with a 15.5% improvement in "Fear" over CORECT and notable gains in differentiating similar emotions like "Anger" (3.5%) and "Disgust" (5.81%) compared to SDT. These results highlight the model's effectiveness in addressing class imbalance while maintaining consistent performance across both major and minority emotion categories.

Overall, the results demonstrate the effectiveness of our unimodal-driven distillation and SDMoE strategy which enhances the student's ability to absorb structured knowledge, while balancing modality-specific and cross-modal features, especially in fine-grained emotional distinctions.

## 4.5 Ablation Studies

To investigate the effectiveness of each component within SUMMER, we conduct ablation studies on both the IEMOCAP and MELD datasets. The results are represented in Table 3 and Table 4.

**Guidelines for Teacher Model Selection.** To assess the effectiveness of the proposed teacher model, our experiment with various combinations of text, audio, and visual modalities, using the original attention mechanism model. As shown in Table 3, the text modality consistently outper-

Table 3: Ablation studies with different modality settings on IEMOCAP and MELD.

| Modality | IEMOCAP | | MELD | |
|---|---|---|---|---|
| | ACC | w-F1 | ACC | w-F1 |
| Text | 69.57 | 69.73 | 66.49 | 65.32 |
| Audio | 67.37 | 67.18 | 55.78 | 55.47 |
| Visual | 66.20 | 66.28 | 53.89 | 53.43 |
| Text+Audio | 71.18 | 70.83 | 67.55 | 66.58 |
| Text+Visual | 69.80 | 69.51 | 67.54 | 66.41 |
| Audio+Visual | 68.05 | 67.49 | 59.01 | 58.33 |
| Text+Audio+Visual | **71.62** | **71.18** | **67.71** | **66.61** |

Table 4: Ablation studies of key components on IEMOCAP and MELD.

| Module | | | IEMOCAP | | MELD | |
|---|---|---|---|---|---|---|
| SDMoE | HCMF | IKD | ACC | w-F1 | ACC | w-F1 |
| ✓ | ✗ | ✗ | 76.52 | 76.64 | 67.43 | 68.57 |
| ✗ | ✓ | ✗ | 76.15 | 75.43 | 67.83 | 68.24 |
| ✗ | ✗ | ✓ | 77.48 | 76.86 | 68.39 | 69.52 |
| ✓ | ✓ | ✗ | 77.82 | 78.13 | 68.17 | 69.04 |
| ✗ | ✓ | ✓ | 77.95 | 77.94 | 68.42 | 69.21 |
| ✓ | ✗ | ✓ | 78.54 | 78.64 | 68.57 | 69.33 |
| ✓ | ✓ | ✓ | **79.11** | **78.95** | **68.78** | **69.81** |

forms others in multimodal emotion recognition, prompting its selection as the teacher model in our framework. While combining text with other modalities offers marginal performance gains, the added complexity and risk of overfitting make unimodal teacher models a more efficient choice.

**Effectiveness of SDMoE modules.** In our ablation study, we replaced SDMoE in SUMMER with the MoE module, as shown in Table 4. Results consistently reveal a performance decline across emotion categories when SDMoE is removed. Moreover, pre-training the teacher model with SD-MoE (Figure 4) shows significant improvements over previous benchmarks. This is attributed to SDMoE's ability to dynamically adjust attention weights and resource allocation, which reduces redundant information with local token-wise interactions, ultimately boosting overall performance.

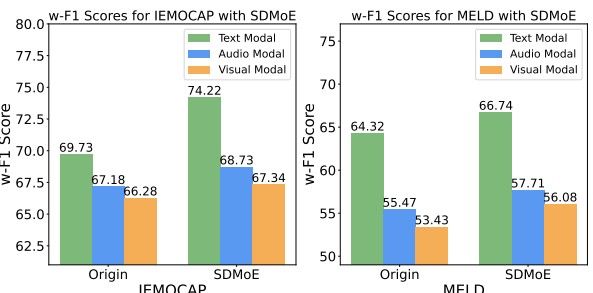

Figure 4: Performance of the SDMoE module across various modalities on the IEMOCAP and MELD datasets.

Figure 5: The Trend Visualization of HCMF Module Loss Functions.

**Impact of HCMF.** To evaluate the HCMF module, we conducted ablation experiments by replacing it with a self-attention mechanism. This led to a noticeable performance decline, confirming that HCMF outperforms static fusion strategies in integrating multimodal information which enhances the model's ability to learn high-level semantic relationships between modalities. Additionally, as shown in Figure 5, we found that introducing residual structures made model training smoother and improved convergence.

**Interactive knowledge Distillation.** As shown in Table 4, the novel interactive distillation achieves the best performance in single ablation experiments, guiding the student model with frozen teacher representations and enhancing its ability to integrate complex inter-modal relationships. Moreover, soft labels preserve relational information between categories better than hard labels, improving generalization and performance. While KL divergence further helps the student model capture subtle inter-class differences, stabilizing training and mitigating gradient conflicts from modality heterogeneity. The detailed experimental setup is discussed in A.2.

**Error Analysis.** In the SUMMER framework, the teacher model excels at capturing fine-grained features in unimodal settings, while the student model benefits from multimodal fusion, offering more generalized yet robust performance. Despite slightly lower results in specific categories, the student model remains strong overall. The underperformance in the "Sad" category may be due

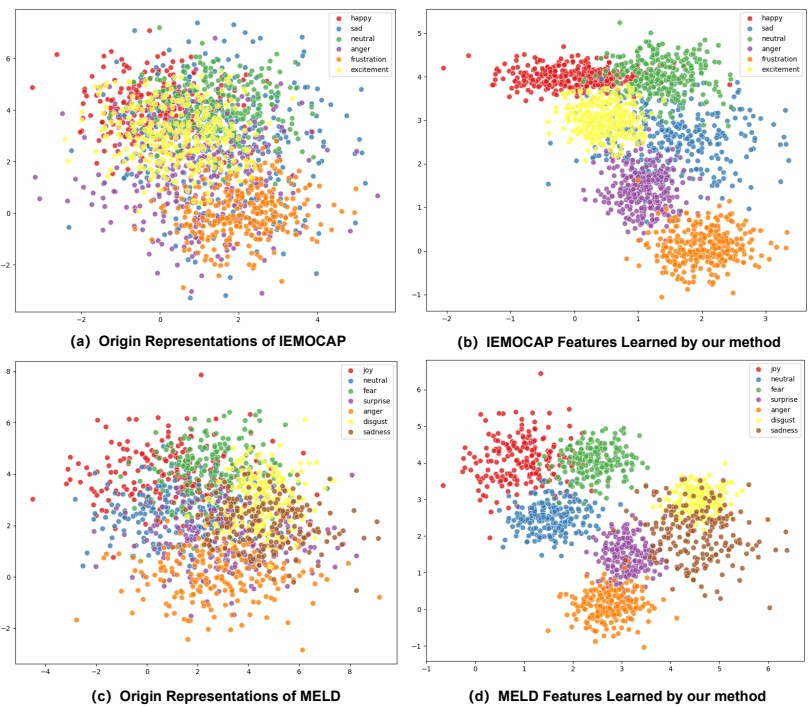

Figure 6: Visualization of features for MERC on the IEMOCAP and MELD datasets. Each point corresponds to an utterance, with colors denoting different emotions. (a) Original features from the IEMOCAP dataset. (b) Features learned by our method on the IEMOCAP dataset. (c) Original features from the MELD dataset. (d) Features learned by our method on the MELD dataset.

to multimodal conflicts, overlapping emotional boundaries (e.g., sadness and frustration), and data imbalance. Addressing these challenges is crucial to improving multimodal emotion recognition.

### 4.6 MULTI-MODAL REPRESENTATION VISUALIZATION

To visually assess the performance of our method, we applied t-SNE to project the high-dimensional multimodal features into a two-dimensional space (Figure 6). The visualization results indicate that while there is still slight overlap between similar emotions (such as "happy" and "excited"), the separation between emotion categories is quite distinct. Notably, SUMMER enhances the clustering of emotion categories, reducing the mixing of closely related emotions and strengthening the distinction between neutral and other emotions. Additionally, the SUMMER model demonstrates greater robustness in integrating multimodal features, allowing it to capture subtle emotional variations more accurately, especially in the presence of data noise and blurred emotional boundaries.

## 5 CONCLUSION

In this work, we propose SUMMER framework for Multimodal Emotion Recognition in Conversations, effectively integrating heterogeneous modalities through a Sparse Dynamic Mixture of Experts for local token-wise interaction and a global Mixture of Experts for context modeling. By employing a novel retrograde distillation method where a unimodal teacher guides a multimodal student model, SUMMER mitigates gradient conflicts and enhances inter-modal relationship learning. Experiments on IEMOCAP and MELD datasets show that SUMMER outperforms state-of-the-art methods, improving recognition of both majority and minority emotion classes, and highlighting its robustness in MERC tasks.

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

# A APPENDIX

## A.1 VISUAL FEATURE EXTRACTION

In conversational analysis, facial expressions serve as crucial indicators of emotional changes in speakers. However, existing studies predominantly employ 3D-CNNs to directly process video streams, a method that introduces several challenges. Video streams often contain a substantial amount of redundant or minimally varying information, as adjacent frames typically exhibit minimal differences, resulting in highly similar extracted features. This not only increases computational load and noise but also complicates the alignment with other modal information.

To address these challenges, we propose an improved visual feature extraction method, termed LENet$_{(3D)}$. Initially, we apply a keyframe extraction strategy, sampling video frames at intervals $N_s = \frac{Frames}{10}$ of 10 frames, denoted as $u_i^v$. Subsequently, facial landmarks in the video frames are detected and aligned using a Multi-task Cascaded Convolutional Network (MTCNN), ensuring the precision of facial region alignment $c_i^v = \text{MTCNN}(u_i^v), i \in \{1, 2, ..., N_s\}$. This process yields a continuous, aligned video stream that serves as input to the 3D-CNN.

We utilize a pre-trained 3D-CNN model fine-tuned on the VGGFace2 dataset, specifically adapted for facial feature extraction tasks. The aligned face video segments are passed through the 3D-CNN, where we extract spatio-temporal feature vectors from intermediate layers rather than the final output layer. Finally, a DialogueRNN network is employed to model the temporal dynamics of both the speaker's emotional states and visual information. The extracted features are reduced to 256 dimensions via a fully connected layer to facilitate further analysis.

## A.2 MODEL DISTRIBUTIONS

In this experiment, we used a text-based teacher model to guide the learning process of the student model, and the results demonstrate significant improvements in the student model's performance. We computed and visualized the feature distribution of the model outputs to further validate the effectiveness of this approach.

As shown in Figure 7. Initially, the student model's feature distribution was more dispersed compared to the well-structured distribution of the teacher model, particularly due to the inherent heterogeneity in multimodal data. However, as the student model learned from the teacher, its distribution gradually converged towards that of the teacher model, showing a clear alignment in the learned feature space. This convergence indicates that the teacher model effectively transfers knowledge, guiding the student model to capture more refined and meaningful features.

By comparing the feature distributions at different stages of training, we observed that the teacher model not only enhances the student model's ability to learn from text but also improves the overall integration of multimodal data. The text-based teacher model proves to be instrumental in resolving challenges of multimodal learning, particularly in cross-modal feature representation.

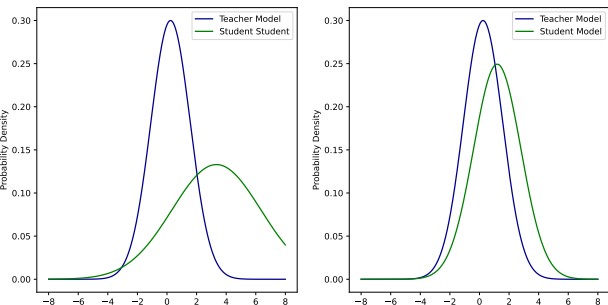

Figure 7: Visualization of distributions of the student model and teacher model.

