# OpenReview forum: "Unimodal-driven Distillation in Multimodal Emotion Recognition with Dynamic Fusion"
_ICLR.cc/2025/Conference — ICLR 2025 Conference Withdrawn Submission_

### Official Review · Reviewer_RdVP · 2024-10-20

**Soundness:** 2
**Presentation:** 1
**Contribution:** 2
**Rating:** 3
**Confidence:** 4

**Summary:**

This paper proposes SUMMER, grounded in attention mechanism and knowledge distillation techniques, which facilitates dynamic interactive fusion of multimodal representations. The experiments validate the effectiveness of the method.

**Strengths:**

1. Several strategies are combined to enhance the performance of MERC.

**Weaknesses:**

1. The motivation is not clearly stated. In line 53, the authors claim that if the model overemphasizes earlier positive expressions, it may make incorrect predictions. The authors just use "if" to state the limitation of existing methods and do not tell why and how existing methods might overemphasize earlier positive expressions.
2. The writing of the paper needs to be improved. Notations are not clear. (E3 $W_g$) Equations are not correct (E4,5,6). Typos in figures and paper should be revised carefully (Fig2, 3(a), 6).
3. Why do you choose $2\sigma$ in Equation 3? The details should be explained.
4. More recent baselines are needed to validate the superiority of the model. It seems that some methods outperform SUMMER in MELD.

**Questions:**

See Weaknesses.

---

### Official Review · Reviewer_Sn2A · 2024-10-29

**Soundness:** 3
**Presentation:** 2
**Contribution:** 2
**Rating:** 5
**Confidence:** 4

**Summary:**

This paper addresses Multimodal Emotion Recognition in Conversations (MERC) with a heterogeneous multimodal integration method called SUMMER based on attention mechanism and knowledge distillation techniques. Specifically, it consists of the Unimodal Teacher Model, Unified Multimodal Student Model, and Interactive Knowledge Distillation. In the Unified Multimodal Student Model, unimodal encoders first extract features from text, audio, and visual inputs, then sparse dynamic MoEs fuse unimodal context information, and finally hierarchical cross-modal fusion module conducts multimodal fusion.

**Strengths:**

# Good results #
- The method achieves SOTA performance on IEMOCAP(6-ways)
- The multimodal student model improves performance effectively
# Method #
- A well-designed sparse dynamic MoEs that fuse unimodal context information
- A hierarchical cross-modal fusion module conducts multimodal fusion

**Weaknesses:**

- The whole framework is built upon the unimodal encoders, thus the method is mainly dependent on them. An evaluation of them is absence.

- The motivation of Sparse Dynamic MoE (SDMoE) and Hierarchical Cross-Modal Fusion (HCMF) are unclear.

- "Unimodal Reconstruction" in the text is inconsistent with the one in Figure 2. Still, the framework is described by three modules in Figure 2 while four modules in the text.

- The loss weights in Eq.(15) are unclear in the experiments.

- The comparisons should include recent LLM-based methods like InstructERC.

- The code is unavailable.

**Questions:**

refer to the weaknesses.

---

### Official Review · Reviewer_8KxT · 2024-11-01

**Soundness:** 3
**Presentation:** 4
**Contribution:** 2
**Rating:** 3
**Confidence:** 4

**Summary:**

This paper presents the SUMMER framework for Multimodal Emotion Recognition in Conversations (MERC), using Sparse Dynamic Mixture of Experts (SDMoE) and Hierarchical Cross-Modal Fusion (HCMF) to enhance multimodal representation and fusion across text, audio, and visual cues. A retrograde distillation method allows a unimodal teacher to guide the multimodal student model, improving fusion and reducing gradient conflicts. SUMMER achieves notable gains on two datasets, outperforming state-of-the-art methods, especially in recognizing minority and semantically similar emotions.

**Strengths:**

1. Detailed explanation of model components: The methodology section in Chapter 3 provides a relatively thorough explanation of each component of the model. Figures 2 and 3 offer clear and intuitive visual aids that help clarify the proposed architecture and its components.
2. Improved performance and error analysis: The proposed method demonstrates performance improvements over state-of-the-art methods, and the paper provides a useful error analysis, highlighting its effectiveness in multimodal emotion recognition in conversations.

**Weaknesses:**

1. Lack of some justification: The introduction part does not adequately explain why the transition to a Mixture of Experts (MoE) model is necessary for addressing the MERC task. There are multiple model architectures and methods applicable to MERC, and MoE is only one of them. The limited justification for choosing MoE may leave readers unclear about the reasoning behind its selection over other architectures. Additionally, the citations provided are not comprehensive; for instance, only a few prior works are referenced in the discussion of MoE-related studies (line 76), and the discussion of Knowledge Distillation (KD) methods (line 139) lacks references to recent advancements in KD, with citations stopping at 2021.
2. Unclear motivation: The paper does not clearly articulate why KD is necessary or advantageous for his task and model. The motivation and background for employing KD are not well established, which weakens the foundation for introducing this technique. Furthermore, the third stated contribution of the paper overlaps with the first two contributions and does not provide enough unique value to warrant being listed as a separate contribution.
3. Issues with details: There are inconsistencies in some parts of the paper. For example, in line 73, the text states “the correct label is excited,” while the label shown in Figure 1 is “excitement.” Additionally, the description, “but dynamic changes in facial expressions and vocal tone might mislead the model to classify it as anger,” lacks a sufficient explanation, making it hard for readers to understand the scenario being described. In the results section (line 411), the text mentions that the proposed method surpasses baselines like CHFusion, particularly in minority classes such as “excitement.” However, Table 1 does not provide any class-specific results for CHFusion.

**Questions:**

1. Could the authors clarify the motivation behind selecting the Mixture of Experts (MoE) and Knowledge Distillation (KD) approaches for the MERC task, and provide additional references or comparisons to alternative architectures commonly used in MERC?
2. In the results and example sections, could the authors clarify why no detailed results for CHFusion are presented for specific emotion classes in Table 1, despite its mention in the text?
3. Additionally, would the authors consider testing on more datasets and providing further experimental analysis?

---

### Official Review · Reviewer_uQfJ · 2024-11-02

**Soundness:** 3
**Presentation:** 2
**Contribution:** 2
**Rating:** 5
**Confidence:** 5

**Summary:**

This paper proposes a sparse dynamic expert mixture and hierarchical cross-modal fusion method to enhance local key marker selection and improve global context understanding, thereby refining heterogeneous modal information to achieve more effective multi-modal fusion. An inverse distillation strategy is introduced, in which a single-modality driven teacher model guides a multi-modal student model, standardizing and solving the fusion disorientation problem in multi-modal learning. Tests on the public MERC dataset show that the SUMMER framework consistently outperforms existing state-of-the-art methods, achieving significant progress in identifying a small number of semantically similar emotions in the MERC task.

**Strengths:**

The results of the experiment are promising.

**Weaknesses:**

1. The authors mentioned in the introduction that existing MERC methods still face challenges such as low modality association efficiency. However, the example (1) cited emphasizes the risk of focusing on the local context and ignoring key emotional cues.
2. In Figure 2, the discourse-speaker embedding belongs to the Sparse Dynamic MoE (SDMOE) module, while the paper writing belongs to the unimodal reconstruction module.
3. In Figure 3 (b), Global MoE belongs to the hierarchical cross-modal fusion (HCMF) module, but in the paper, it belongs to the Sparse Dynamic MoE (SDMOE) module.
4. In section 3.5 of the paper, it is mentioned that the authors set up a single-text modality teacher model to drive multimodal learning, but the connection between the two cannot be seen in Figure 2.
5. There is a Residual part in the HCMF part in Figure 2 which is not introduced in the paper.
6. The classifier part in Figure 2 is not introduced in the paper.
7. Parametric experiments are missing.
8. Model complexity needs to be evaluated.
9. It is advised to present a direct comparison with the reported results in more recent literature.

**Questions:**

1. The research motivation and method logic of the paper need to be carefully sorted out.
2. The experiment needs to be strengthened.

---

### Official Review · Reviewer_PN8r · 2024-11-02

**Soundness:** 2
**Presentation:** 2
**Contribution:** 2
**Rating:** 3
**Confidence:** 4

**Summary:**

In the MERC task, the cross-modal redundant information and the gradient conflict caused by modal heterogeneity limit the effectiveness of multimodal fusion representations. This paper proposes a method called SUMMER to facilitate dynamic interactive fusion of multimodal representations. Experiments on two datasets demonstrate SUMMER outperforms existing state-of-the-art methods.

**Strengths:**

The use of retrograde distillation in the MERC task is novel, and the authors provide ablation experiments that demonstrate its effectiveness.

**Weaknesses:**

**The figures and writing of this paper are confusing.**
(1) Fig.3(b) indicates that HCMF has four inputs. However, in Fig.2, HCMF_{text} has only one input, and HCMF_{***} has three inputs.
(2) Section 3.4 introduces SDME, while Section 3.5 discusses HCMF. However, the Global MOE, as a part of HCMF, is detailed in Section 3.4.

**The reproducibility of this work is poor.**
(1) The proposed method has a large number of hyperparameters, such as loss weights (line 349), distillation temperature (line 249), and the soft-label parameter (line 343). More importantly, the authors do not provide specific values or relevant ablation studies.

**The experimental section is insufficient.**
(1) There is a lack of comparison with other distillation-based MER methods [1,2,3].
(2) The authors also lack corresponding analyses for two contributions. For instance, regarding the first contribution, there is a lack of visual analysis to demonstrate how the proposed method enhances local key token selection and improves understanding of the global context.

**Other minor issues**
(1) Some results for comparison methods are reported as zero, which is uncommon.
(2) In line 196, the authors claim to have proposed LFNet. In reality, it is a common sampling strategy in action recognition and dynamic facial expression recognition tasks.
[1] Multi-Task Momentum Distillation for Multimodal Sentiment Analysis. TAFFC2023.
[2] Decoupled Multimodal Distilling for Emotion Recognition. CVPR2023.
[3] Muti-modal Emotion Recognition via Hierarchical Knowledge Distillation. TMM2024.

**Questions:**

Please refer to the weaknesses.

---

### Note · Authors · 2024-11-20

I have read and agree with the venue's withdrawal policy on behalf of myself and my co-authors.